# Stochastic Control for Bayesian Neural Network Training

**DOI:** 10.3390/e24081097

**Published:** 2022-08-09

**Authors:** Ludwig Winkler, César Ojeda, Manfred Opper

**Affiliations:** 1Machine Learning Group, Technische Universität Berlin, 10623 Berlin, Germany; 2Artificial Intelligence Group, Technische Universität Berlin, 10623 Berlin, Germany; 3Centre for Systems Modelling and Quantitative Biomedicine, University of Birmingham, Birmingham B15 2TT, UK

**Keywords:** Bayesian inference, Bayesian neural networks, learning

## Abstract

In this paper, we propose to leverage the Bayesian uncertainty information encoded in parameter distributions to inform the learning procedure for Bayesian models. We derive a first principle stochastic differential equation for the training dynamics of the mean and uncertainty parameter in the variational distributions. On the basis of the derived Bayesian stochastic differential equation, we apply the methodology of stochastic optimal control on the variational parameters to obtain individually controlled learning rates. We show that the resulting optimizer, StochControlSGD, is significantly more robust to large learning rates and can adaptively and individually control the learning rates of the variational parameters. The evolution of the control suggests separate and distinct dynamical behaviours in the training regimes for the mean and uncertainty parameters in Bayesian neural networks.

## 1. Introduction

Deep Bayesian neural networks (BNNs) aim to leverage the advantages of two different methodologies. First, in recent years, deep representations have been incredibly successful in fields as diverse as computer vision, speech recognition and natural language processing [1,2,3]. Much of the success, however, revolves around prediction accuracy. Second, Bayesian methodologies are required to obtain an estimate of model uncertainty, a crucial feature that allows deep neural networks to tackle risk assessment to create informed model decisions. The role of model uncertainty in the training procedure of BNNs, however, remains unaddressed; the present investigation seeks to exploit the model uncertainty in Bayesian neural networks for the development of new learning algorithms.

For the training of BNNs, the approximate posterior over the model parameters is obtained via a maximization of the variational lower bound.

Such a posterior introduces a form of uncertainty in the parameters which is different than that injected by random batches of data. In this investigation, we seek to exploit both the data uncertainty (aleatoric) and the model uncertainty (epistemic) to solve a control problem aimed at maximizing the evidence lower bound (ELBO), where the control parameters gauge the dynamics of the gradient during descent.

The contributions of our work are threefold,

We provide a derivation of the stochastic differential equation on a first principle basis that governs the evolution of the parameters in variational distributions trained with variational inference and we decompose the uncertainty of the gradients into their aleatoric and epistemic components.We derive a stochastic optimal control optimization algorithm which incorporates the uncertainty in the gradients to optimally control the learning rates for each variational parameter.The evolution of the control exhibits distinct dynamical behaviour and demonstrates different fluctuation and dissipation regimes for the variational mean and uncertainty parameters.

Section 1 offers an introduction to the topic. In Section 2, we provide an overview over probabilistic models and Bayesian neural networks. Section 3 details the derivation of the stochastic differential equation governing the dynamics of the frequentist and variational parameters. Subsequently, we derive a stochastic optimal control algorithm in Section 4 on the basis of the dynamics of the variational parameters. Finally, Section 5 summarizes experiments undertaken and the performance of the stochastic optimal control optimizer, as well as the distinct behaviour of the control parameters.

## 2. Variational Inference for Bayesian Neural Networks

For a training dataset D, the Bayesian formulation of a neural network places a posterior distribution p(θ|D) and a prior p(θ) on each of its parameters θ. The quintessential task in Bayesian inference is to compute the posterior p(θ|D) according to Bayes’ rule:(1)$$\begin{array}{c} p\left(\theta |D\right)=\frac{p(D|\theta )p(\theta )}{p(D)}. \end{array}$$
Given a likelihood function p(D|θ) and the parameter prior p(θ), we can make predictions by marginalizing out over the parameters. For the most common application of supervised learning with label *y* and data *x*, $D={\left\{y_{m},x_{m}\right\}}_{m=1}^{N},x\in X,y\in Y$, this gives us
(2)$$\begin{array}{c} p(y|x)=\int p(y|x,\theta )p(\theta |D)d\theta . \end{array}$$
where p(y|x,θ) is the likelihood function of the output *y* given the input *x* and the posterior parameter distribution p(θ|D). For highly parameterized models, the inference of the posterior distribution p(θ|D) requires the computation of a high dimensional integral which is numerically intractable to compute for most complex models as they can easily have millions of parameters.

There are two main approaches for inferring the posterior distribution p(θ|D) in Bayesian neural networks: sampling from the posterior distribution in proportion to the data likelihood and prior [4], and variational inference, which optimizes a bound on the evidence and approximates the true posterior with a tractable distribution q(θ|ϕ)≈p(θ|D) with the variational parameters ϕ, [5].

Our approach focuses on the variational inference formulation, which scales well to large data regimes as the bound is amenable to gradient-based optimization schemes [6]. Variational inference infers the posterior distribution by optimizing the Kullback–Leibler divergence between the true posterior p(θ|D) and a variational distribution q(θ|ϕ). The important detail is that the variational distribution is assumed to be independent of the data D, which makes the solution approximate yet tractable. The optimization problem is then
(3)$$\begin{array}{c} \arg\min_{\phi }\;\mathrm{KL}\left[q(\theta |\phi )||p(\theta |D)\right]=\arg\min_{\phi }\;-E_{q(\theta |\phi )}\left[\log p(D|\theta )\right]+\mathrm{KL}\left[q(\theta |\phi )||p(\theta )\right]. \end{array}$$

We are, thus, left to optimize the ELBO, which is derived in full in Appendix A, in the form $E_{q(\theta |\phi )}\left[\log p(D|\theta )\right]-\mathrm{KL}\left[q(\theta |\phi )||p(\theta )\right]$ as a surrogate loss function. The ELBO is optimized numerically through gradient descent algorithms, which bring their own set of challenges with respective to gradient step size, directional sensitivity and exploding and vanishing gradients. We propose a stochastic optimal control algorithm for gradient descent optimization which controls the learning rate for every variational parameter ϕ based on the local surface of the Kullback–Leibler divergence.

For the remainder of this paper, we assume that the variational distribution q(θ|ϕ) for each parameter θ follows an independent normal or Laplace distribution with the location of the distribution μ and the scale σ as the variational parameters ϕ={μ,σ} of the parameter θ. Since the scale parameter σ is constrained to be positive, we employ an additional reparameterization σ=log(exp(ρ)+1) which allows us to compute derivatives for ρ during optimization while keeping σ strictly positive.

### 2.1. Stochastic Differential Equations for Frequentist Models

During optimization, the model parameters follow a dynamical process; in the following section, we show how it is possible to approximate this dynamic as an SDE; we start with a frequentist version where no distribution is imposed on the parameters (as in the BNN), and the stochasticity is injected by the dataset and samples from therein. Given a probabilistic model $p(y_{m}|x_{m},\theta ):X\to Y$ with a set of scalar parameters θ∈R, the input $x_{m}\in X$ and output $y_{m}\in Y$, we compute the derivative of a scalar loss function $L_{m}:Y\times Y\to R$ to obtain the derivative with respect to each parameter $\partial_{\theta }L_{m}$ in the probabilistic model for a single data point. Gradient descent requires us to calculate the derivative of the loss over the entire training dataset D at each iteration $\partial_{\theta }L=1/\left|D\right|\sum_{i=1}^{\left|D\right|}L_{i}$. This gradient, has an associated variance:(4)$$\sigma_{D}=V_{D}\left[\partial_{\theta }L\right]=\frac{1}{\left|D\right|}\sum_{i\in D}{\left(\partial L_{i}-\partial_{\theta }L\right)}^{2}.$$
The computational cost of calculating gradients over entire training datasets is prohibitively expensive, which has favoured the use of mini-batch sampled gradients. Now, a mini-batch with M≪|D| data points is sampled [7]. The assumption is that a mini-batch is computationally tractable while providing a representative sample of the training dataset to compute a sufficiently good gradient on. We denote $D_{m}$ as a single data sample and $D_{M}$ as the mini-batch sample. The sampling of the mini-batches introduces stochasticity into the gradient estimation. The first and second moments, denoted as $E\left[\cdot \right]$ and $V\left[\cdot \right]$, for each scalar parameter θ of the mini-batch gradients are:(5)$$\begin{array}{cc} \partial_{\theta }L_{M} & =\frac{1}{M}\sum_{m=1}^{M}\partial_{\theta }L_{m} \end{array}$$(6)$$\begin{array}{c} \partial_{\theta }L=E_{M∼p(D)}\left[\partial_{\theta }L_{M}\right] \end{array}$$(7)$$\begin{array}{c} V_{M}\left[\partial_{\theta }L_{M}\right]=\frac{1}{M}\sum_{i\in D}{\left(\partial L_{i}-\partial_{\theta }L\right)}^{2} \end{array}$$(8)$$\begin{array}{c} V_{M}\left[\partial_{\theta }L_{M}\right]\propto \frac{1}{M}\sigma_{D} \end{array}$$
It is easy to see that we can decrease the variance in the gradient estimation by increasing the size of the mini-batch *M*. The change in the parameters $\Delta \theta_{t}=\theta_{t+1}-\theta_{t}$ in gradient based optimization consequentially follows a noisy estimate of the true gradient $\partial_{\theta }L$ which is distributed according to the first- and second-order moments in (4) and (5). The central limit theorem implies that the derivatives are distributed along a Gaussian distribution, $\partial_{\theta }L_{M}∼N\left(\partial_{\theta }L,\frac{1}{M}\sigma_{D}\right)$ [8]. Given the distribution of the gradients, the evolution of the parameter through time with the learning rate η can be approximated by:(9)$$\begin{array}{c} \theta_{t+1}=\theta_{t}-\eta \partial_{\theta }L_{M} \end{array}$$(10)$$\begin{array}{c} \Delta \theta_{t}=-\eta \partial_{\theta }L+\eta \sqrt{\frac{\sigma_{D}}{M}}\;\epsilon \;;\epsilon ∼N\left(0,1\right) \end{array}$$

This formulation of the parameter dynamics during training has strong similarities with the Euler–Maruyama discretization of an Ito drift–diffusion process. Indeeed, for an SDE with drift $b(\theta_{t})$ and diffusion σ(θ):(11)$$d\theta_{t}=b\left(\theta_{t}\right)dt+\sigma \left(\theta_{t}\right)dW_{t}$$
we have the associated Eurler–Maruyama discretization:(12)$$\theta_{t+1}=\theta_{t}+b\left(\theta_{t}\right)\Delta t\sqrt{\Delta t}\sigma \left(\theta \right)\epsilon .$$
We proceed by setting η≡Δt, $b\left(\theta_{t}\right)\equiv \partial_{\theta }L$ and $\sigma \left(\theta_{t}\right)\equiv \sqrt{\eta /M\sigma_{D}}$ to denote equivalency, as further described in [8,9,10]. This modification allows the use of stochastic analysis to Ito drift-diffusion processes. See [11] for a more thorough discussion on the relationship of the learning rate and the diffusion of SGD). If we additionally consider the learning in the infinitesimal limit of η→0, we arrive at a formulation for the instantaneous change in time which is given by
(13)$$d\theta_{t}=-\partial_{\theta }Ldt+\sqrt{\eta /M\sigma_{D}}dW_{t}$$
which is a stochastic differential equation, where $dW_{t}$ is a Wiener process that originates from the limit applied to $\sqrt{\eta }\epsilon$ [12]. We can, thus, conclude that the change in the parameters $\theta_{t}$, for an infinitesimal small learning rate η, follows a stochastic differential equation in the form of an Ito drift–diffusion process over time in which the sampling of the mini-batches contributes the diffusion [12].

### 2.2. Stochastic Differential Equations for Bayesian Models

In BNN models, each scalar parameter θ is modelled by a univariate distribution θ∼q(θ|ϕ). The use of the distribution q(θ|ϕ) extends the loss L to the form of the ELBO which is additive in the mini-batch samples *m* and has a closed form regularization term (the Kullback–Leibler divergence between posterior p(θ|D) and prior p(θ)), the derivation of which can be found in Appendix A. Not only do we choose data samples at random, but, concurrently, we sample the parameter θ from the distribution q(θ|ϕ) following the reparametrization trick. The parameter θ is thus a random variable itself. Consequentially, the derivative $\partial_{\theta }L_{m}$ for a single data sample *m* will exhibit randomness originating both from the randomly sampled mini-batches and the stochasticity of the sampled parameters from the variational distribution.

The uncertainty of the parameter derivative $\partial_{\theta }L$ can be decomposed into the aleatoric and the epistemic uncertainty. The aleatoric uncertainty arises from the variance in the data and is irreducible, whereas the epistemic uncertainty arises from the uncertainty of the parameter θ and can be reduced to zero, since, in principle, the parameters can be sampled θ.

Employing the tractable univariate variational distribution q(θ|ϕ) to achieve a scalable optimization, for a derivative $\partial_{\theta }L_{m}$ which is dependent on the random parameter θ and the randomly chosen data sample *m*, we can decompose the uncertainty of $\partial_{\theta }L$ into a sum of the data uncertainty and the parameter uncertainty, which follows from the law of total variance [13]:(14)$$\begin{array}{cc} V\left[\partial_{\theta }L\right]= & \underset{\mathrm{Aleatoric}\;\mathrm{Uncertainty}}{\underbrace{V_{p(D_{M})}\left[E_{q(\theta |\phi )}\left[\partial_{\theta }L_{m}|D_{m}\right]\right]}}+\underset{\mathrm{Epistemic}\;\mathrm{Uncertainty}}{\underbrace{E_{p(D_{M})}\left[V_{q(\theta |\phi )}\left[\partial_{\theta }L_{m}|D_{m}\right]\right]}}. \end{array}$$
In effect, we draw samples twice in BNNs to obtain ‘per data sample per variational sample’ derivatives: data samples for the mini-batch and parameter samples θ from the variational distribution. Aleatoric uncertainty first computes the expectancy over the ‘variationally sampled’ derivatives per data sample $D_{m}$ and subsequently computes variance over the mini-batch $D_{M}$. Epistemic uncertainty first computes the variance over the ‘variationally sampled’ gradients and, finally, computes the expected derivative over the mini-batch $D_{M}$.

It is important over which source of randomness the variance is computed in the uncertainty decomposition. The first term, $V\left[E\left[\partial_{\theta }L_{m}|D_{m}\right]\right]$, represents the aleatoric uncertainty and measures the data uncertainty. It measures how much the average gradient varies over the dataset. The second term, $E\left[V\left[\partial_{\theta }L_{m}|D_{m}\right]\right]$, is called the epistemic uncertainty and measures the uncertainty originating from the model parameter distribution. For the epistemic uncertainty, the variance is computed over the source of parameter uncertainty and averaged over the data samples. In BNNs this is explicitly modelled through the use of distributions for every parameter θ. Frequentist models exhibit only aleatoric uncertainty, as the variance over the deterministic gradients in the epistemic uncertainty evaluates to zero.

For a univariate variational distribution θ∼q(θ|ϕ), we can now formulate the stochastic differential equation (SDE) that governs the dynamics of the variational parameters ϕ={μ,σ}.

The first modification, with respect to the SDE of a frequentist model in Equation (10) is that, for every parameter θ in the frequentist model, we have, in fact, two separate variational parameters ϕ={μ,σ} in the Bayesian model, corresponding to the mean and scale of the variational distribution from which we sample θ. We, thus, have the two differential equations for the variational parameters {μ,σ},
(15)$$\begin{array}{cc} d\mu_{t} & =-E\left[\partial_{\mu }L\right]dt+V{\left[\partial_{\mu }L\right]}^{\frac{1}{2}}dW_{t} \end{array}$$
(16)$$\begin{array}{cc} d\sigma_{t} & =-E\left[\partial_{\sigma }L\right]dt+V{\left[\partial_{\sigma }L\right]}^{\frac{1}{2}}dW_{t}^{*}. \end{array}$$
in which $d\sigma_{t}$ has a separate Wiener process $dW_{t}^{*}$ due to the externalized noise in the reparameterization, the details of which can be checked up upon in the Appendix C. The second modification is the separation of uncertainty, given that we have the additional source of uncertainty from the distribution q(θ|ϕ). We can, thus, employ the uncertainty decomposition to obtain
(17)$$\begin{array}{c} d\mu_{t}=-E\left[\partial_{\mu }L\right]dt+{\left(V\left[E\left[\partial_{\mu }L_{m}|D_{m}\right]\right]+E\left[V\left[\partial_{\mu }L_{m}|D_{m}\right]\right]\right)}^{\frac{1}{2}}dW_{t} \end{array}$$
(18)$$\begin{array}{c} d\sigma_{t}=-E\left[\partial_{\sigma }L\right]dt+{\left(V\left[E\left[\partial_{\sigma }L_{m}|D_{m}\right]\right]+E\left[V\left[\partial_{\sigma }L_{m}|D_{m}\right]\right]\right)}^{\frac{1}{2}}dW_{t}^{*} \end{array}$$

We can now see that the only difference in the SDEs that govern the training dynamics in frequentist and Bayesian models is the added epistemic uncertainty in the diffusion term of the Bayesian stochastic differential equation.

Figure 1 exemplifies the different terms in the Bayesian stochastic differential equation and how uncertainty in stochastic gradient descent for a variational distribution can be decomposed for a toy example in one dimension. The details of the derivation of the Bayesian stochastic differential equation can be followed up in Appendix C.

## 3. Stochastic Control for Learning Rates

Having derived and characterized the training dynamics of the variational parameters $x\equiv \{\mu_{t},\sigma_{t}\}$ on a first principle basis, we now construct our proposed stochastic optimal control algorithm for BNNs. Our approximation methodology relies on the limit η→0. We first introduce a new control variable that respects the limit, namely the learning rate adjustment to the training, an additional adaptive diagonal control matrix *U*, which leads to a full SDE for the dynamics of training as:(19)$$\begin{array}{c} {\dot{x}}_{t}=-U\;E\left[\nabla_{x}L\right]\;dt+U\sqrt{\eta }V{\left[\nabla_{x}L\right]}^{\frac{1}{2}}dW_{t}\; \end{array}$$
which is an Ito drift-diffusion process, where both the drift and the diffusion are controlled by the diagonal control matrix *U* and where the diffusion term is estimated from the variance of the gradients. We essentially scale it on a per-parameter basis with the control matrix *U*. We clip the individual control parameters $U_{i}$ on the diagonal of *U* to the range $U_{i}\in \left[0,1\right]$ bounding the step size to $\eta U_{i}\in \left[0,\eta \right]$. We posed our problem as follows: if we have the gradients $\nabla_{x}L$, how do we choose the policy for adjusting the control parameter *U* to minimize the loss at the end of the training? Essentially:(20)$$\min_{u}E\left[L\left(X_{T}\right)\right]\;\mathrm{subject}\;\mathrm{to}\;(19)$$
provided that *X* follows Equation (Equation 19). The general optimal control formalism requires us to minimize the cost *C* for the optimal control parameter *U*, accumulated over time $t\in [t_{0},t_{1}]$, and the final cost $C(x_{t},U,t_{1})$, under the constraint of the dynamics ${\dot{x}}_{t}$.

### 3.1. Simplifiying the Loss

It is known that the loss surface L of deep neural network architectures is highly non-linear, which makes global optimization nearly impossible. In a similar way to [14,15], we therefore approximate the loss surface locally with a quadratic function of the form
(21)$$\begin{array}{c} g\left(x_{t}\right)=\frac{1}{2}{\left(x_{t}-b\right)}^{⊤}A\left(x_{t}-b\right).\; \end{array}$$
The quadratic approximation as seen in Figure 2 forfeits the global loss surface for a local approximation in which the respective optimal quantities can be computed optimally in the sense of the local approximation. This simplification is chosen such that a tractable stochastic optimal control algorithm can be derived. Intuitively, given a local quadratic approximation of the loss surface, the offset parameter *b* denotes the optimum of the quadratic approximation $g(x_{t})$, whereas the curvature *A* denotes how flat or steep the loss surface is locally.

Consequently, we want to move the variational parameters $\left\{\mu_{t},\rho_{t}\right\}$ in the observable state vector $x_{t}$ as close as possible to this local optimum which coincides with the offset parameter *b*.

The curvature *A* and the offset parameter *b* of the local quadratic approximation of the loss surface can be conveniently calculated via ordinary least squares with the gradient relation (see Appendix D for details)
(22)$$\begin{array}{c} \nabla_{\phi }L∼\nabla_{x}g\left(x_{t}\right)=A\left(x_{t}-b\right). \end{array}$$

We maintain running averages of the gradients and the parameters to prevent abrupt changes in the control. The quadratic approximation of the loss surface is maintained for each parameter distribution in the BNN architectures.

### 3.2. Our Control Problem

Taking inspiration from the local quadratic approximation $g(x_{t})$, we wish to minimize the distance of the observable state variables $x_{t}$ to the optimum *b* of the quadratic approximation g(x). We introduce an auxilliary variable *L* which allows us to simplify the classical control problem that requires the solution for the Hamiltonian–Jacobi–Bellman equation Appendix E.

It is known, by definition, that $M_{ij}=\left(x_{i}-b_{i}\right)\left(x_{j}-b_{j}\right)$ is a stochastic variable. We can obtain a relationship between *L* and the approximation of the error *g*:(23)$$\begin{array}{cc} g(x_{t}) & =\frac{1}{2}M_{11}A_{11}+M_{12}A_{12}+\frac{1}{2}M_{22}A_{22} \end{array}$$

We make use of Ito’s lemma, detailed in Appendix B, to obtain the dynamics of the error dM and define the diffusion matrix $D=\eta V\left[\nabla_{x_{t}}L\right]$ which gives us,
$$\begin{array}{c} dM=\left(-\nabla_{x}M\;U\;\nabla_{x}L+\frac{\eta }{2}Tr\left[DU^{T}\;\nabla_{x}^{2}M\;U\right]\right)dt+\nabla_{x}M^{T}U\sqrt{\eta V\left[\nabla_{x}L\right]}dW \end{array}$$
which is again an Ito drift-diffusion process, and for which we provide the relevant gradient calculations in Appendix D. With the intention of separating the drift and evaluating the matrix derivatives, the details of which are in the Appendix D, we obtain
(24)$$\begin{array}{c} f(M,U,t)=-(UAM+MAU)+\eta \;UDU \end{array}$$

The dynamics of the error f(M,U,t) denote the drift of the Ito drift-diffusion process dM and represent the average dynamics of the error function over time, given the dynamics $dx_{t}$ of the parameters $x_{t}={\left(\mu_{t},\rho_{t}\right)}^{⊤}$. The task which we want to achieve is to minimize the loss in (Equation 23) in such a way that we arrive at the optimum after the control period $t\in [t_{0},t_{1}]$.
(25)$$\begin{array}{c} C\left(M,U,t_{1}\right)=\frac{1}{2}\int_{t_{0}}^{t_{1}}\mathrm{Tr}[A\dot{M}]dt. \end{array}$$
where *A* is the curvature of the local approximation $g(x_{t})$. The motivation of this formulation is that *M* measures the distance of the state vector $x_{t}$ to the local optimum *b* in the quadratic approximation scaled by the curvature *A*. Thus, minimizing the distance *M* at each time step is equivalent to minimizing the entire cost *C*. The full derivation can be found in Appendix E.

The optimization of the final cost *C* can be solved by minimizing the cost of $\mathrm{Tr}[A\dot{M}]$, which, in turn, minimizes *C*. Taking the derivative of $\frac{1}{2}\mathrm{Tr}\left[A\dot{M}\right]$ with respect to the individual control parameters $U_{ii}$ and setting it to zero gives us
(26)$$\begin{array}{c} U^{\prime }={\left(A\circ D\right)}^{-1}\frac{\mathrm{Diag}\left[AMA\right]}{\eta } \end{array}$$
where $U^{\prime }={\left[U_{11},U_{22}\right]}^{⊤}$ is a vector with the corresponding control parameters, ∘ is the Hadamard product and Diag[·] extracts the diagonal elements of a matrix For indefinite matrices *A*, we project $U^{\prime }$ onto the eigenvector corresponding to the positive eigenvalue to ensure that the optimality condition is met [16]. The full derivation can be reviewed in Appendix E.

We compute the control parameter $U^{\prime }$ jointly for the variational parameters {μ,σ}, which results in the matrices *A*, *D*, *M* to be in $R^{2\times 2}$. The inversion of the 2×2 matrices can be performed analytically, as detailed at the end of Appendix F. Comparing the operations required per parameter in ADAM (addition, subtraction, division etc.) and those in StochControlSGD (mostly 2 × 2 matrix multiplications and analytical inversions), we arrive at an approximately 2.5× increase in computations for StochControlSGD compared to ADAM. It is important to note that ADAM has to be applied to both variational parameters independently, whereas StochControlSGD computes the control parameters jointly, thus saving computation.

The StochControlSGD algorithm is detailed in its entirety in Algorithm 1.
**Algorithm 1:** StochControlSGD
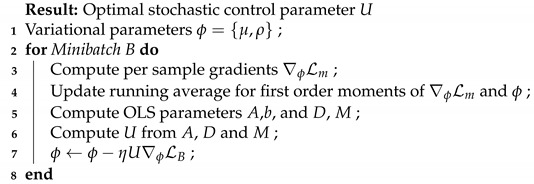


## 4. Experiments

We evaluate the proposed stochastic optimal control SGD, which we abbreviate as StochControlSGD, on the MNIST [17], FashionMNIST [18] and CIFAR10 [19] datasets. In Table 1, we compare the final performance of StochControlSGD with the performance of ADAM, controlled SGD (cSGD), SGD and SGD with cosine learning rate scheduling, as proposed by [15].

Learning rate scheduling was chosen as the cosine annealing, where the initial learning rate was chosen as $10^{-1}$ and was decreased to $10^{-5}$. The experimental setup is detailed in Appendix G.

ADAM provides a strong baseline for the frequentist models when the learning rate is chosen to be appropriately small. Following the notion of learning rate scheduling, we initialized the learning of both cSGD and StochControlSGD as η=0.5 and $u_{0}=1.0$. Both cSGD and StochControlSGD are able to adaptively and individually set their control parameters over the course of optimization.

Additionally, we plot the convergence of the ADAM, cSGD and StochControlSGD in Figure 3. The results are portrayed more concisely in Figure 4, for which five runs for each learning rate and each optimizer are combined in a boxplot format.

In contrast to cSGD and StochControlSGD, ADAM does not have the ability to modify the a priori chosen learning rate η. Coupled with the first- and second-order moments from which the surrogate gradient is computed, ADAM is sensitive to the large learning rate with significantly worsening performance for learning rates at η=0.5 and η=0.1. The larger learning rates do not pose a problem for the optimal control optimizers cSGD and StochControlSGD, as they can adaptively and individually control their learning rates. We consider only optimizers which rely on the gradient information to accelerate the gradient descent and forego learning rate scheduling algorithms which incorporate performance information, such as learning rate schedulers which decrease the learning rate if a performance plateau is detected.

Among the optimal control optimizers, StochControlSGD provides tighter bounds on the lower and upper performance while offering a higher performance. Especially on the CIFAR10 dataset in Figure 3, StochControlSGD improves upon cSGD with better absolute performance and less variation between the largest learning rate of η=0.5 and the smallest learning rate η=0.01. Furthermore, it can be seen that the performance of StochControlSGD and cSGD improve with larger learning rates. As can be seen in Figure 3, the performance of the largest learning rate of η=0.5 is, in fact, its best performance, whereas it is the worst performance for ADAM.

The direct comparison of ADAM with StochControlSGD connects to recent work carried out by [20] on the fundamental optimization of deep Bayesian models with gradient optimization algorithms developed for frequentist models. The methodology of BNNs is limited in the amount of relevant information in the uncertainty with respect to the learning optimization due to its reliance on normal priors. Modern frequentist deep neural networks rely on custom layer architectures, such as BatchNorm [21], with additional data augmentation schemes, which have no clear Bayesian interpretation, raising additional questions on the applicability of porting frequentist ideas, such as layer designs, in deep neural networks, to their Bayesian formulations.

### Behaviour of Control Parameter

The evolution of the control parameter *U* allows insight into the descent and fluctuation behaviour of the variational parameters $\mu_{t}$ and $\rho_{t}$ with respect to the ELBO. More specifically, it allows us to shed some light onto the dynamics between the data log likelihood and the KL divergence.

The data loglikelihood aims at minimizing the uncertainty parameter $\rho_{t}$ of each variational distribution as much as possible. The gradients of KL divergence, in turn, prioritize an uncertainty parameter which corresponds to the prior which we chose as N(0,I). The relative weighting of the data log likelihood and KL divergence with respect to the number of samples in the ELBO heavily favours the gradients of the data log likelihood during the descent phase for large datasets. As the gradients of the KL divergence are independent of the data by definition, the importance of their gradients increases proportionally to the diminishing gradients of the converging data log likelihood.

The uncertainty parameters were initialized to $\rho_{0}=-6.9$ in all our experiments which allows the BNN to increase the uncertainty of select parameters if the KL divergence dominates the gradients of the specific parameter in question. The intuition is that deep neural networks, in fact, only use few weights [22], and, thus, the uncertainty parameters can be maximized by the KL divergence for parameters for which the gradients of the KL divergence are stronger than the gradients originating from the data log likelihood.

We can observe this behaviour in Figure 5, where the median control parameters of $\mu_{t}$ decrease quickly alongside the control parameters for the uncertainty parameter $\rho_{t}$. However, as the data loglikelihood converges, the median control parameter of the uncertainty parameter is increased as the relative importance of the gradients originating from the data loglikelihood decreases and the gradients from the KL divergence dominate.

This indicates two different dynamical regimes in the optimization of the uncertainty parameter of the variational distribution. The mean control parameter remains small during the descent and fluctuation dynamics whereas the uncertainty control is, in fact, increased by the stochastic control optimization algorithm in the fluctuation phase.

## 5. Related Work

The authors of [15] derived an optimal control algorithm for frequentist models which incorporated the variance into the learning rate scheduling. In [23], it was argued that instead of decreasing the learning in the dissipation phase of the optimization, the batch size should be increased to reduce the uncertainty in the gradients. The authors of [24] and [25] examined adaptive learning rate schemes for changing loss surfaces. The idea of a priori cyclical scaling in the learning rates was pioneered in [26].

The use of the reparameterization of the Gaussian variational distribution in deep Bayesian neural networks to arrive at a scalable optimization algorithm based on variational inference was proposed in [27]. The authors of [28] examined the behaviour of DropOut [29] as an approximate Bayesian inference. The authors of [30] demonstrated that the dropout rate could be learned as an approximate uncertainty parameter.

## 6. Conclusions

We have examined the potential for incorporating Bayesian uncertainty information directly into a learning algorithm. For this, we derived the SDEs for variational parameters on a first principle basis. With both aleatoric and epistemic uncertainty present in the optimization process, we decomposed the diffusion parameter of the SDE into its data and parameter uncertainties.

Having identified the underlying dynamics of the variational parameters during optimization, we proceeded to formulate a stochastic optimal control algorithm for Bayesian models which was able to incorporate the Bayesian uncertainty information into an adaptive and selective learning rate schedule. An analysis of the control parameters indicated separate dynamical behaviours during optimization of the mean and uncertainty parameters. This can be investigated further to examine the dynamics of the ELBO as a loss function for other probabilistic models.

## Figures and Tables

**Figure 1 entropy-24-01097-f001:**
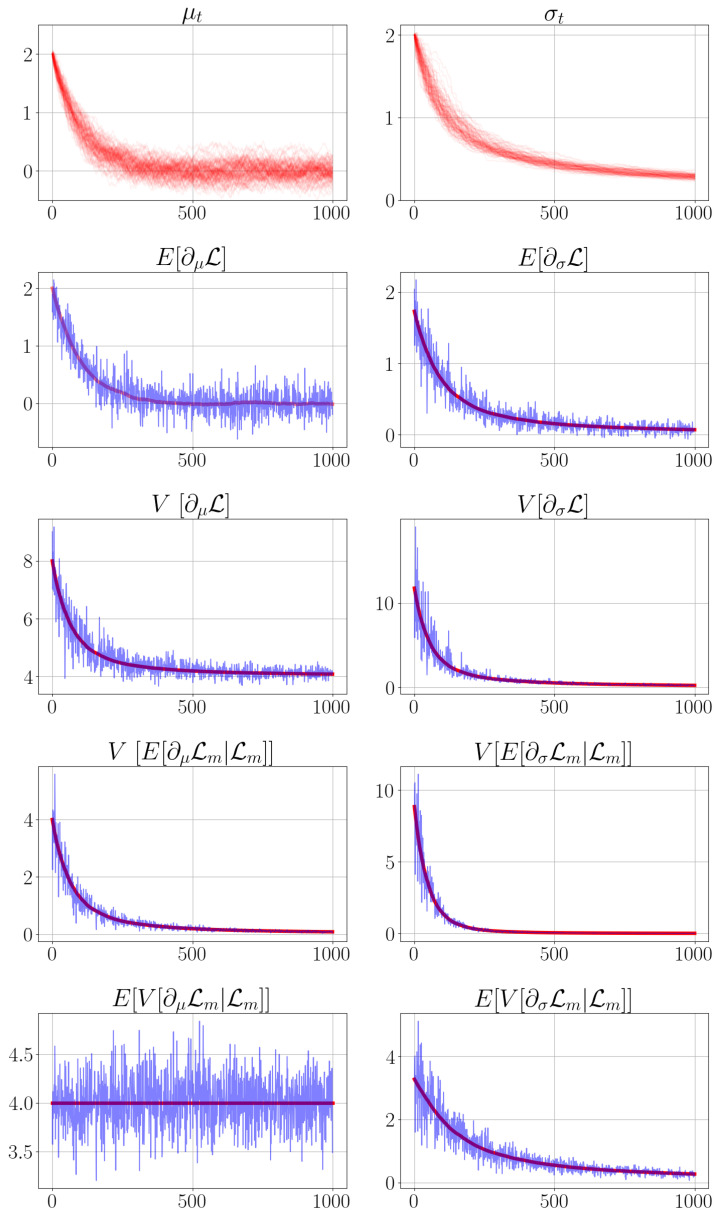
The components of the stochastic differential equation for the variational parameters $\mu_{t}$ and $\sigma_{t}$ over time. The empirical drift and diffusion estimates shown in blue are unbiased estimates of the true analytically derived drift and diffusion terms. The loss was $L=\frac{1}{2}{\left(\theta_{t}-b\right)}^{2}$ where *b* was sampled randomly from b∈{−2,+2} to simulate aleatoric uncertainty. The aleatoric uncertainty from the data in the gradients remains constant whereas the epistemic uncertainty from the parameter distribution is reduced to zero.

**Figure 2 entropy-24-01097-f002:**
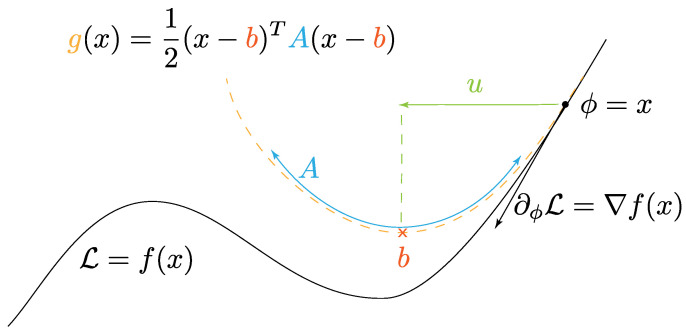
A one-dimensional illustration of how the optimal stochastic control *u* is determined from the gradient and parameter information. The parameters ϕ and their gradient information $\partial_{\phi }L$ are used to estimate the curvature *A* and offset *b* for the quadratic approximation *g* through which the optimal control parameter *u* is determined. In our experiments with Bayesian neural networks, each parameter θ has two variational parameters ϕ={μ,σ}, such that $A\in R^{2x2}$ and $b\in R^{2}$.

**Figure 3 entropy-24-01097-f003:**
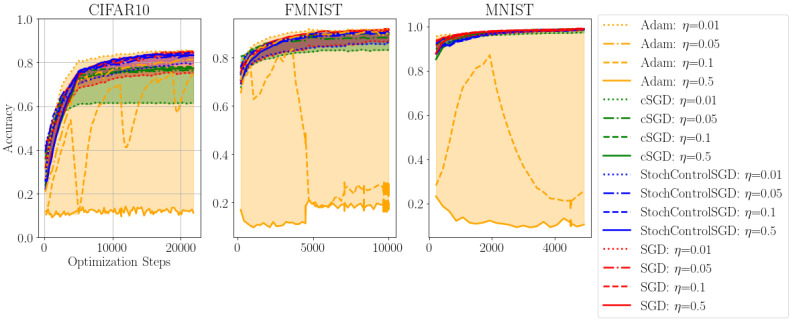
Comparison of StochControlSGD with SGD, controlled SGD and ADAM. StochcontrolSGD offers very robust performance over varying learning rates.

**Figure 4 entropy-24-01097-f004:**
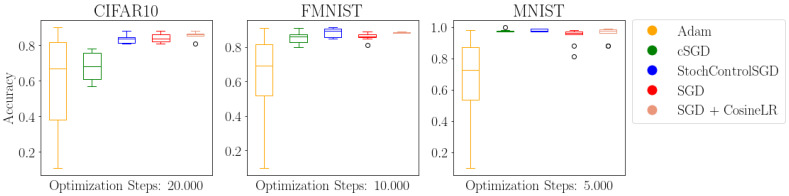
Combined performance of the optimizers over different learning rates. StochControlSGD provides reliable performance over a wide range of learning rates without the necessity of hyperparameter tuning.

**Figure 5 entropy-24-01097-f005:**
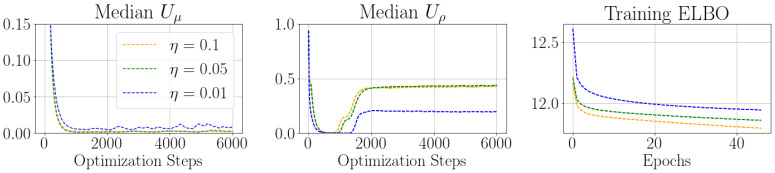
The median control parameter over time plotted with the Training ELBO which is used to compute the gradients for a BNN which was trained on Fashion MNIST.

**Table 1 entropy-24-01097-t001:** Test accuracy on the MNIST, FMNIST and CIFAR10 datasets. We abbreviate StochControlSGD as scSGD, and the SGD with cosine learning rate scheduling as LRSGD, for notational brevity. The best performing optimization algorithm per data set is denoted in bold.

	MNIST					FMNIST					CIFAR10				
	SGD	ADAM	cSGD	scSGD	LRSGD	SGD	ADAM	cSGD	scSGD	LRSGD	SGD	ADAM	cSGD	scSGD	LRSGD
NN	0.959	**0.987**	0.961	/	0.985	0.818	**0.890**	0.851	/	0.878	0.461	**0.512**	0.432	/	0.499
CNN	0.989	**0.993**	0.981	/	0.990	0.904	**0.918**	0.912	/	0.907	0.853	**0.865**	0.857	/	0.855
BNN (Normal)	0.956	0.963	0.970	**0.971**	0.069	0.865	0.870	0.876	**0.900**	0.900	0.441	0.442	0.451	**0.471**	0.462
CBNN (Normal)	0.982	0.988	0.982	**0.990**	0.989	0.869	0.914	0.903	**0.921**	0.915	0.615	**0.854**	0.836	0.853	0.801
BNN (Laplace)	0.976	**0.978**	0.974	0.977	0.975	0.890	0.875	**0.903**	0.901	0.9	**0.501**	0.452	0.461	0.479	0.500
CBNN (Laplace)	0.989	0.987	0.985	**0.991**	0.989	0.899	0.916	0.907	**0.918**	0.912	0.627	0.857	0.829	**0.857**	0.853

## Data Availability

The training data and code base used in this study are available upon reasonable request from the authors.

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
