# Peer review of "Stochastic Control for Bayesian Neural Network Training"

_entropy, 2022, doi:10.3390/e24081097_

Round 1

Reviewer 1 Report

SUMMARY

The authors propose a method to optimise Bayesian neural networks based on stochastic optimal control. The paper is well-written and the algorithm presented offers some insight into the training dynamics of Bayesian neural networks. As such, I recommend acceptance with the following points the authors may wish to consider.

MAJOR POINTS

1. In the empirical evaluation it may also be worth investigating uncertainty metrics of the optimised Bayesian neural networks in addition to accuracy given that this is the main motivation for using BNNs in place of standard deep networks. I leave this decision to the authors.

2. It would be great if the code for the paper could be released open-source as this would help greatly in clarifying some of the experimental details. For example which approximate BNN implementation was used? Presumably stochastic gradient Langevin dynamics based on reference 35 in the manuscript? [1].

3. It may be worth providing an estimate of the computational overhead saved in applying the stochastic control algorithm when compared with hyperparameter tuning of the learning rate in e.g. Adam. As the reviewer understands, the advantage of having an adaptive learning rate scheduler in practice may be that it avoids computationally expensive hyperparameter tuning routines.

MINOR POINTS

1. It would be great if "Bayesian" was capitalised in "Bayesian neural networks" in the keywords section.

2. The first paragraph seems to imply that Bayesian methodologies are strictly necessary if one requires an estimate of model uncertainty? In terms of approaches such as ensemble methods are the authors assuming that all models constituting the ensemble can receive weight under the prior?

3. In the first paragraph "Bayesian networks" could be confused for the probabilistic graphical model instead of the "neural" network.

4. Is the proposed methodology applicable to all BNNs that optimise a variational lower bound?

5. It might be an idea to define the BNN acronym early in the manuscript and then using this convention throughout.

6. The acronym for ELBO appears to be introduced before the evidence lower bound is first mentioned.

7. At the end of the introduction paragraph, is it correct to demarcate the parameters into variational and frequentist parameters or would it be preferable to refer to the parameters as the network weights?

8. Typo, missing full stop at the end of equation 1.

9. In section 2,  it may be preferable to refer to the likelihood as the likelihood function as opposed to the likelihood distribution in order to empahsise that it is a function of the model parameters i.e. the weights of the neural network [2].

10. It may be worth emphasising in section 2 that the high dimensional integral is intractable i.e. non-analytic and that it is impractical to compute NUMERICALLY by Monte Carlo methods.

11. Perhaps the term "marginal data probability" could be changed to have consistency in terminology throughout the manuscript?

12. "Amenable" may be a more appropriate word choice to "amicable".

13. Some punctuation may be missing in the equations of appendix A in the derivation of the KL divergence.

14. It may be worth providing the derivation of the ELBO with Jensen's inequality in appendix A e.g. as in https://www.cs.princeton.edu/courses/archive/fall11/cos597C/lectures/variational-inference-i.pdf. Going from A2 to A3, it might be worth emphasising that rewriting the approximate posterior as q(\theta | D) instead of q(\theta | \phi) above is a notational mechanism to highlight the dependence on D as opposed to the variational parameters \phi.

15. Line 61, typo, "gradient descent algorithms".

16. Line 65 may need revising.

17. Section 2.1 would recommend referring to the data set as the training set for consistency.

18. Full stop missing at the end of equation 4.

19. Equation 8 might be worth using a different symbol to indicate proportionality so as not to be confused with, "is distributed according to".

20. Towards the bottom of page 3, when mentioning that the mini-batch gradients are distributed according to a Gaussian distribution by the Central Limit Theorem, should the gradient derivative have a subscript M?

21. In equation 10, it might be clearer if the learning rate \eta was placed in front of the derivative in the first term.

22. Top of page 4 should Euler Murayama be hyphenated?

23. Extraneous comma below equation 12? What is \eta set to? Is the proportionality symbol the most appropriate here?

24. Line 84, typo in "chose".

25. It may be worth further elaborting on the precise meaning of aleatoric uncertainty in relation to the variance in sampling minibatches viz-a-viz the definition corresponding to noise in the data-generation mechanism i.e. measurement noise.

26. Typo in capitalisation of uncertainty in footnote 2 on page 4.

27. Typo in figure 1, \rho_t?

28. Typo on the top of page 13 in infinitessimal.

29. Typo line 128.

30. In appendix F, it may be worth providing the long and tedious algebra for completeness!

31. In the references section, Adam was accepted at ICLR 2015 [3].

32. Typo, missing capitalisation of Bayesian in the titles of some references.

33. Line 190, might it help for "Gaussian priors" to be referred to as Normal priors for consistency?

REFERENCES

[1] Welling and Teh, Bayesian Learning via Stochastic Gradient Langevin Dynamics. ICML, 2011.

[2] Information Theory, Inference and Learning Algorithms, David MacKay, Cambridge University Press, 2004.

[3] Kingma and Ba, ADAM: A Method for Stochastic Approximation, ICLR, 2015.

Author Response

Major Points

Q1 In the empirical evaluation it may also be worth investigating uncertainty metrics of the optimised Bayesian neural networks in addition to accuracy given that this is the main motivation for using BNNs in place of standard deep networks. I leave this decision to the authors.

A1 Thank you for highlighting this analysis. We intend to do an in-depth paper on the uncertainty calibration while leveraging the uncertainty information during training.

Q2 It would be great if the code for the paper could be released open-source as this would help greatly in clarifying some of the experimental details. For example which approximate BNN implementation was used? Presumably stochastic gradient Langevin dynamics based on reference 35 in the manuscript?

A2 The code will be released upon acceptance as a footnote in the final manuscript. We parameterize the BNN via the 'Weight Uncertainty in Neural Networks' approach by Blundell et al. Every neural network parameter (used in the linear projections in each layer of the NNs) is replaced by variational distributions with the reparameterization trick at the heart of Variational AutoEncoders by Kingma et al among others.

This is a variational inference approach in contrast with sampling based approaches such as SGLD in Welling et al.

Q3 It may be worth providing an estimate of the computational overhead saved in applying the stochastic control algorithm when compared with hyperparameter tuning of the learning rate in e.g. Adam. As the reviewer understands, the advantage of having an adaptive learning rate scheduler in practice may be that it avoids computationally expensive hyperparameter tuning routines.

A3 We have added a paragraph beneath the main algorithm detailing the extra cost of running StochControlSGD compared to Adam. In essen it is approximately 2.5 times slower where the most expensive computations are done in the estimation of the loss surface of the joint variational parameters {\mu, \sigma}.

Minor Points

  1. Has been addressed.
  2. We assume that ensembles are approximate Bayesian methods as they draw a limited number of samples from the true posterior distribution.
  3. Has been corrected
  4. Yes, the methodology is applicable to all Bayesian neural networks that are trained via maximization of the variational lower bound.
  5. It has been addressed in the main text.
  6. This has been fixed.
  7. In variational inference BNN, the optimizable parameters are the variational parameters location and scale in the variational distribution from which the weights are sampled. In frequentist neural networks, one uses the term parameter/weight. Thus it would be correct to say that the network weights are sampled from a distribution parameterized by the variational parameters in BNNs.
  8. Has been corrected.
  9. Has been corrected.
  10. Has been added.
  11. 'marginal data probability' has been changed to evidence to reflect the precise term.
  12. Thank you for the suggestion and 'amenable' has been used.
  13. The punctuation has been added to Appendix A.
  14. Thank you for the feedback. We would retain the current derivation starting from the loss function as we think it might be more accessible for people from the optimization community. We have followed your advice highlighting the posterior distribution conditioned on the data D.
  15. Thank you for highlighting the typo.
  16. We have rewritten the sentence around line 65 in the hopes that it is clearer now, what we aim to achieve.
  17. This has been corrected in the new manuscript.
  18. Thank you and it has been corrected.
  19. We have used the LaTex command \propto.
  20. Thank you very much for the very attentive reading. This has been corrected.
  21. This has been corrected.
  22. Our literature has consistently hyphenated the integration scheme. We would stick with the hyphenation. 
  23. We have removed the comma, set \eta equivalent to the step size \Delta t and have used the equivalency sign to denote the move from discreet to continuous.
  24. Has been corrected
  25. TO BE DONE
  26. Thank you has been corrected
  27. Thank you. It has been corrected to \sigma_t.
  28. Has been addressed.
  29. 'the' has been added to address the typo.
  30. The derivation in all its algebraic glory has been added. 
  31. The ADAM paper was correctly cited.
  32. The capitalization has been done. Thank you.
  33. This has been addressed.

Reviewer 2 Report

: In this paper the authords propose to leverage the Bayesian uncertainty information encoded in  the parameter distributions to inform the learning procedure for Bayesian models. The authors derive a  first principle stochastic differential equation for the training dynamics of the mean and uncertainty parameter in the variational distributions. On the basis of the derived Bayesian stochastic differential equation, they apply the methodology of stochastic optimal control on the variational  parameters to obtain individually controlled learning rates. We show that the resulting optimizer,  StochControlSGD, is significantly more robust to large learning rates and can adaptively and  individually control the learning rates of the variational parameters. The evolution of the control  suggests separate and distinct dynamical behaviours in the training regimes for the mean and uncertainty parameters in Bayesian neural networks. 

Although the paper is interesting and well written I would like to see a comparison with the more mainstream back propagation or gradient descent methodologies.

Author Response

Q1 Although the paper is interesting and well written I would like to see a comparison with the more mainstream back propagation or gradient descent methodologies.

We added SGD and SGD with Cosine Learning Rate Scheduling to the experimental setup.

In our impression, ADAM and SGD with Learning Rate Scheduling represent the baselines as well as most used optimizers used in the deep learning community.

Round 2

Reviewer 2 Report

The revised version is OK